# Phytoplankton Morpho-Functional Trait Variability along Coastal Environmental Gradients

**DOI:** 10.3390/microorganisms9122477

**Published:** 2021-11-30

**Authors:** Sirpa Lehtinen, Sanna Suikkanen, Heidi Hällfors, Jarno Tuimala, Harri Kuosa

**Affiliations:** 1Marine Research Centre, Finnish Environment Institute (SYKE), 00790 Helsinki, Finland; sanna.suikkanen@syke.fi (S.S.); heidi.hallfors@syke.fi (H.H.); harri.kuosa@syke.fi (H.K.); 2Independent Researcher, 00790 Helsinki, Finland; jtuimala@gmail.com

**Keywords:** morpho-functional traits, phytoplankton, eutrophication, climate change, nitrogen-fixation, mixotrophy, motility, buoyancy, size, harmfulness

## Abstract

We utilized the trait-based approach in a novel way to examine how specific phytoplankton traits are related to physical features connected to global change, water quality features connected to catchment change, and nutrient availability connected to nutrient loading. For the analyses, we used summertime monitoring data originating from the coastal northern Baltic Sea and generalized additive mixed modeling (GAMM). Of the physical features connected to global climate change, temperature was the most important affecting several studied traits. Nitrogen-fixing, buoyant, non-motile, and autotrophic phytoplankton, as well as harmful cyanobacteria, benefited from a higher temperature. Salinity and stratification did not have clear effects on the traits. Water transparency, which in the Baltic Sea is connected to catchment change, had a mostly negative relation to the studied traits. Harmfulness was negatively correlated with transparency, while the share of non-harmful and large-sized phytoplankton was positively related to it. We used nutrient loading source type and total phosphorus (TP) as proxies for nutrient availability connected to anthropogenic eutrophication. The nutrient loading source type did not relate to any of the traits. Our result showing that N-fixing was not related to TP is discussed. The regionality analysis demonstrated that traits should be calculated in both absolute terms (biomass) and proportions (share of total biomass) to get a better view of community changes and to potentially supplement the environmental status assessments.

## 1. Introduction

Phytoplankton communities are composed of primary producers, and they form the base of the pelagic food webs. However, phytoplankton is not a homogenous group; but instead, there is great diversity among phytoplankton taxa, as well as in the morpho-functional traits of the taxa. Certain phytoplankton traits are universal for almost all aquatic environments, even though the systematic group or the species carrying them may vary between the environments. For example, the ability to fix atmospheric inorganic nitrogen (N) is carried in marine systems, e.g., by the cyanobacterium *Trichodesmium* and symbiotic cyanobacterial species [1], in certain brackish water ecosystems mainly by *Nodularia spumigena*, while in certain lower-salinity brackish water and freshwater ecosystems the same trait is carried by *Aphanizomenon* or *Dolichospermum*, and in the benthos by *Anabaena* and several other cyanobacterial genera. Since functional trait-based approaches can transcend habitat-specific or local taxonomy, it has been suggested that aquatic scientists could use functional trait-based approaches as a common framework to enhance knowledge sharing between freshwater, marine, benthic, and pelagic ecologists [2]. The trait-based approach differs markedly from the taxonomic approach in that traits are universal for almost all aquatic environments, while species composition varies between different ecosystems. However, information on the species identity of a specimen is often more than the sum of its’ trait information. Thus, the trait-based approach does not replace the taxonomic approach; instead, the approaches complement each other [3,4,5].

Functional traits are usually defined as genotypic (through genes carried by an organism) and phenotypic (through the observable expression of genes) characteristics that mediate growth, reproduction, and survival of an organism [6] and determine its fitness for ambient biotic and abiotic conditions [3]. Many traits are included in the currently available trait data tables as qualitative traits expressing species’ potential to carry a certain trait (e.g., a binary trait expressing potential for mixotrophy based on available literature), but also some quantitative and realized trait data are available (e.g., numerical values for light-dependent growth rate measured in situ or in the laboratory) [2]. There are already several open access trait data tables available for phytoplankton [7,8,9]. Even though regional as well as global functional trait data tables are being supplemented continuously, they are not comprehensive concerning either taxa or traits. The development of trait data tables is an ongoing process since there is constant development of measuring techniques that will make it possible to continuously supplement tables with new traits or with more detailed information.

Some basic phytoplankton traits include the abovementioned potential to fix atmospheric N, as well as motility, buoyancy, mixotrophy, cell size, and harmfulness. The potential to fix atmospheric N is present in diazotrophic heterocystouscyanobacterial species that are a component of the phytoplankton globally in marine, brackish, and freshwater ecosystems [10,11]. N-fixation gives a competitive advantage to these species in situations of N limitation [12,13]. Still, N-fixation in marine systems, including estuaries, coastal seas, and oceanic waters, is regulated not only by N limitation but also by complex interactions of chemical, biotic, and physical factors [14]. In addition, it is known that a high amount of N-fixing cyanobacteria does not always ensure active N-fixation, and N-fixing cyanobacteria may also dominate in cases when N is not limiting because they prefer ammonium as their inorganic N source [15]. In addition to diazotrophic cyanobacteria themselves, N-fixation is assumed to support the concurrent phytoplankton community, since up to 50% of the fixed N may be released by the cyanobacteria within hours [16,17]. Thus, the ability of certain cyanobacteria to fix N may support the dynamics of the whole plankton community during N-limitation [18].

Motility and buoyancy allow phytoplankton cells to regulate their vertical position in the water column in order to select for favorable growth conditions [19,20,21]. Motility is a phenotypic cross-systematic characteristic, and it means the ability of flagellated cells to swim three-dimensionally. It is present in all flagellated species, such as dinoflagellates, cryptophytes, euglenoids, prymnesiophytes, most chrysophytes, and some chlorophytes. In our study, buoyancy as a definable trait is a genotypic trait present in some cyanobacteria that can control their vertical position in the water column with gas vesicles [22] and in the genus *Botryococcus* (Chlorophyta, Trebouxiophyceae), species of which may produce large quantities of hydrocarbons that are excreted and retained extracellularly to maintain the colonies’ buoyancy [23].

Mixotrophy is the ability of an organism to utilize both phototrophy and phagotrophy for their nutrition, i.e., they function both as autotrophs and as heterotrophs [24,25]. Mixotrophy is commonly present in some cryptophytes, prymnesiophytes, dinoflagellates, and chrysophytes. In a phytoplankton community, the amount of mixotrophs reflects the importance of acquiring nutrients via bacteria and other plankton instead of dissolved nutrients. In nutrient-depleted situations, a mixotrophy-based food web may be more productive than a traditional autotrophy-based food web [26].

Cell size is a key trait connected to the surface-to-volume ratio and affects the growth rate, nutrient uptake, and storage capacity, as well as sinking rate [27,28,29]. Small cells are faster growing and do not sink as easily compared to large cells. Organism size also affects the risk of being grazed by certain types of grazers. While microzooplankton feed on smaller-sized phytoplankton, copepods feed on both microzooplankton as well as on ca. >10 µm phytoplankton [30,31,32]. However, as is known, many of the trait-based functions are generalizations. For example, the direct relationship of cell size and sinking rate may be more complicated in real communities [22].

Harmfulness can be related to toxin production, as well as to high biomass of non-toxic phytoplankton causing, e.g., anoxia or mucilage, that negatively affect the environment and human activities. The majority of toxin-producing species are dinoflagellates, but they also include cyanobacteria, diatoms, haptophytes, raphidophytes, dictyochophytes, and pelagophytes [33]. Harmful species can negatively affect other phytoplankton (allelopathy) [34,35,36,37], heterotrophic plankton [38,39], fish [40,41], birds [42], or mammals [43]. The reasons for producing harmful substances may be connected to achieving a competitional advantage over other phytoplankton species, or avoiding grazing, but it has also been speculated that cyanobacteria may produce harmful substances only as a by-product of their metabolism without a specific purpose [44]. The increasing risk of harmful algal blooms (HABs) has been connected to global climate change and anthropogenic eutrophication [45], although the universality of this connection has since been questioned [46].

In the Baltic Sea, phytoplankton community composition and its seasonal succession have mainly been studied using taxonomic approaches [47]. In recent years, the trait-based approach has been introduced to the Baltic Sea phytoplankton studies by one study in which the functional phytoplankton community structure and its drivers were studied [8]. The selected monitoring dataset that we used in our study has not previously been used for trait-based analyses. The aim of our study was to utilize the trait-based approach and natural phytoplankton community data to gain additional knowledge on potential connections between phytoplankton composition and environmental variables. There is a high demand for additional information on these connections, especially for the applied purposes of coastal marine status assessments. The trait-based approach might be well suited for these purposes because it is not tied to local species composition in the same way as the taxonomy-based approach. Unlike the earlier trait-based study using Baltic Sea phytoplankton data [8], we used trait-specific biomasses and shares of total biomass in the analyses. The reason for this is that we focused specifically on certain traits, and not on the overall functional community structure. Furthermore, our approach, i.e., using biomasses and their shares of total phytoplankton biomass, might be more feasible to be adopted into marine management. We specifically examined how certain phytoplankton traits may be linked to different scales of environmental change, specifically changes in physical features due to global change, in water quality features due to catchment change, and in nutrient availability due to nutrient loading. The phytoplankton morpho-functional traits included in our study were the potential for N-fixation, mixotrophy, motility, buoyancy, harmfulness, as well as size. The environmental factors considered were temperature, salinity, total phosphorus concentration (TP), water transparency (Secchi depth), stratification of the water column, and nutrient loading source type. The generalized additive mixed modeling (GAMM) approach was used to consider the hierarchical data structure of our dataset (sea area > water body > sampling station). To our knowledge, similar studies utilizing the trait-based approach have not been published earlier.

We hypothesized that (1) the potential for N-fixation is positively correlated with TP (since N does not limit the growth of N-fixers [13]), (2) the potential for motility or buoyancy (i.e., the ability to select for favorable growth conditions) is more important when water transparency is low and stratification is strong [19,20,21,48], (3) the potential for mixotrophy correlates negatively with factors that support autotrophic growth, i.e., the availability of nutrient resources [26] (TP in our study) and water transparency [49,50], (4) organism size is positively correlated with the availability of nutrient resources (since small cells are more efficient in taking up nutrients which is an advantage in nutrient-poor conditions) and negatively correlated with stratification (since large cells sink faster) [27,28,29], and (5) harmfulness is positively correlated with temperature and nutrient resources (i.e., global climate change and nutrient pollution [45]).

## 2. Materials and Methods

### 2.1. Study Area

Our data originate from the northern part of the Baltic Sea. The Baltic Sea is a large brackish water basin in northern Europe, divided into sub-basins according to the bathymetry [51,52]. The study area covers Finnish coastal areas in the sub-basins of the Bothnian Sea (BS), the shallow Archipelago Sea (AS), and two parts of the Gulf of Finland, i.e., the deeper, more saline western Gulf of Finland (wGF), and the shallower, less saline eastern Gulf of Finland (eGF) (Figure 1).

In the northern Baltic Sea, salinity varies between ca. 1–6.5 psu [51] and phytoplankton communities consist of species of both marine and freshwater origin [53]. In addition to variability in salinity, there is also variability in other environmental conditions of the coastal areas, including loads of organic matter and nutrients [54], as well as in the taxonomic phytoplankton community composition [55]. Thus, monitoring data originating from this area offer a valuable empirical source to study the variability of morpho-functional phytoplankton traits along coastal environmental gradients.

Within the four areas, there were 80 monitoring stations situated in 44 water bodies. Water bodies are clearly distinguishable areas of surface water, and they are an important entity in the EU Water Framework Directive (WFD) [56] in relating water protection to natural hydrological units. For each water body, information on the share of different N and P loading sources is available (river, point, sediment, offshore) [57]. We used this information to categorize water bodies into five different nutrient loading source types (L): (1) N and P mainly from offshore; (2) N from rivers and P from a point source; (3) N from rivers and P from offshore; (4) N and P from rivers; (5) N from rivers and P from the sediment.

### 2.2. Phytoplankton and Environmental Data

We utilized monitoring data collected by the Finnish environmental authorities in 2009–2020 during mid and late summer (1 July–15 September). This is a period following the warming of the water and the development of a strong seasonal thermocline in the surface layer, but before the autumnal mixing of the water column breaks up the thermocline. In the study area, this period is the season of the highest phytoplankton biomass after the spring bloom. Summer is also the period examined for the environmental status assessments required by the WFD.

Phytoplankton samples and environmental data were collected on the same day. Originally, the total number of suitable phytoplankton community samples was 2078, but since there was a significant amount of gaps in the environmental data, the final number of samples included in the analyses was 912. All data are stored in the Finnish national open database OIVA (http://www.syke.fi/en-US/Open_information, accessed 11 November 2021, in Finnish) and the database of the European Marine Observation and Data Network (EMODnet, https://emodnet.ec.europa.eu/en/portals, accessed 11 November 2021).

We selected those phytoplankton samples that were sampled, preserved, stored, microscopically analyzed, and converted to taxon-specific biomass (μg L^−1^ wet weight) results following the guidelines described in the Manual for Marine Monitoring in the COMBINE program of the Helsinki Commission (HELCOM) [58]. Pooled samples were taken with a tube sampler from the surface down to a depth decided according to the Secchi depth (from 0 m to the depth of max twice the Secchi depth). Phytoplankton samples were preserved with acid Lugol’s solution and analyzed using inverted light microscopy [59]. Biovolumes were converted to biomasses (μg L^−1^ wet weight) [60], assuming a density of 1 kg m^−3^.

Heterotrophic taxa, akinetes, heterocytes, cysts, as well as taxa that are mainly benthic or littoral and only occasionally observed in pelagic samples [53] were excluded from the dataset. Single-celled picoplankton and *Synechococcus* were inconsistently enumerated. Their results were also excluded from the dataset since the used monitoring method (light microscopy of Lugol’s-preserved samples [58]) is not suitable for reliable identification of <2 µm-sized single phytoplankton, i.e., it is not possible to separate single-celled picoplankton from single-celled heterotrophic bacteria. Counting results for *Anabaena* were included, since this genus name was earlier also used for the pelagic *Dolichospermum* taxa (nowadays *Anabaena* refers to benthic taxa only [61]).

Environmental variables included surface water temperature, salinity, TP, water transparency (Secchi depth), and stratification (E). For temperature, salinity, and TP, we calculated mean values for the surface layer based on data collected from 0–10 m depth (or if the station was shallower than 10 m, the entire water column). Temperature and salinity data originate from discrete water samples collected using a tube sampler. TP was measured spectrophotometrically [62,63], with a detection limit of 0.01 μM. To determine the stratification index E, densities of the 0–10 m surface layer and near-bottom (deepest measurement 1 m above the bottom) water were calculated from the temperature and salinity of the respective water layers in R [64] using the *rLakeAnalyzer* package [65]. Using these densities and the depth of the deepest temperature and salinity measurement, E was calculated as follows [66]:E = [_σ (bottom) − σ (surface)_] × 1000/depth_deepest TempSal_(1)
where _σ_ = density (kg m^−3^), Temp = temperature (°C), and Sal = salinity (psu).

### 2.3. Morpho-Functional Traits

We assigned the following phytoplankton morpho-functional traits to the monitoring data records a posteriori: N-fixation, mixotrophy, motility, buoyancy, cell size (small/large, average Equivalent Spherical Diameter (aveESD)), and harmfulness (harmful cyanobacteria, harmful eukaryotic phytoplankton). All traits except for the size traits refer to the potential of the function, not whether it was actually expressed in the particular community. Trait values for N-fixation, mixotrophy, motility, and buoyancy were assigned based on the review of the literature and an existing trait data table for the Baltic Sea phytoplankton [8]. Our trait data table, including trait information for the ca. 700 taxa (ca. 2100 different counting units) that were observed in the selected monitoring data, is available as Appendix A.

Harmfulness refers to the potential to cause harm, according to the list of toxin-producing taxa observed in the Baltic Sea [67]. This trait was divided into harmfulness caused by cyanobacteria and harmfulness caused by eukaryotic phytoplankton. Cyanobacteria are the major harmful phytoplankton group in the Baltic Sea. They are a permanent fixture, especially in summertime phytoplankton communities, and they can be considered as a unique entity in our analysis due to their systematic closeness. Other harmful phytoplankton include a wide variety of toxin-producing species [67], which are more enigmatic in their presence. However, their abundance may have commonality with the changing environmental conditions like temperature and nutrients.

In the case that a specimen was identified to a lower taxonomic precision than species-level (e.g., to genus-level), it was generally considered a potential carrier of the trait if one or more of the species in the genus carry the trait. The exceptions were *Snowella* spp., which was not considered to carry the buoyancy trait, since the clear majority of *Snowella* species in the study area do not carry the trait, and *Amphidinium* spp, which was not considered to be harmful since the majority of *Amphidinium* species in the study area are not toxic. *Anabaena* spp. and *Aphanizomenon* spp. were considered harmful since *Dolichospermum lemmermannii* (syn. *Anabaena lemmermannii*) and *Aphanizomenon flosaquae* are listed as harmful [67], and they are very common species in the study area. Furthermore, Prymnesiales, *Prymnesium* cf. *minutum*, and *Chrysochromulina* spp. were considered harmful, since in the order there are several toxic species [67,68], many of which occur in the Baltic Sea [53] and determining them to species or even genus level is in most cases not possible in light microscopical analysis of Lugol’s preserved samples.

Size traits were assigned using two different methods. The first size trait (small/large) indicates if the maximum morphometric measure of the counting unit (cell, colony, coenobium, chain, or 100 µm piece of filament) is less than or equal to 10 µm or if it is larger than that. The other size trait (aveESD) was calculated as follows. First, the number of cells per counting unit was determined. This was done based on information in the dataset, or if lacking, either based on information in Olenina et al. [60] and its annually updated appendix, version 2020, as concerns closely related taxa, similar size classes, or morphologically similar taxa or if not applicable, based on literature. Next, the biovolume per cell was calculated from that of the original counting unit. Then, by assuming a spherical shape for each cell, the radius and, consequently, the diameter for each cell was calculated, arriving at an Equivalent Spherical Diameter (ESD). The total taxon ESD was calculated by using the cell density (i.e., number of cells L^−1^) of each taxon and its ESD. Average sample ESD (aveESD) was calculated by the sum of total taxon ESDs divided by the total cell number in the sample.

For N-fixation, mixotrophy, motility, buoyancy, small/large cell size, and harmfulness, we calculated a sample-specific biomass as the sum of the biomasses of the taxa carrying the trait. The share of each trait-specific biomass of the total sample biomass was calculated based on these values. The biomass and the share of total biomass were also calculated for taxa that did not carry the trait (i.e., non-N-fixing, non-motile, non-buoyant, autotrophic (=non-mixotrophic), and non-harmful taxa). Concerning cell size measured as ESD, aveESD was used in the analyses.

### 2.4. Statistical Analyses

Most of the response variables were not normally or even unimodally or symmetrically distributed and were therefore boxcox-transformed before further analyses [69]. Response variables were modeled using a generalized additive mixed model (GAMM). A single GAMM model was fitted for each response variable, resulting in altogether 27 different models. Temperature, salinity, TP, water transparency (Secchi depth), stratification index E, loading source type L, and sea area were used as explanatory variables in each model. Since the stations where the samples were taken from are nested inside water bodies that are nested inside sea areas, the nested structure was modeled using a random effect in the model. The potential autocorrelation was considered by modeling the time with a continuous autoregressive component in the model. In addition, heteroscedasticity was modeled using variance weights for areas in the model. The results from the GAMM analyses are presented on the (boxcox) transformed scales of the response variables.

Pairwise post-hoc comparisons for different sea areas were carried out using estimated marginal means with Tukey’s adjustment for *p*-values. This method is conceptually similar to Tukey’s HSD post-hoc test with matching results in the case of equal sample sizes.

All statistical analyses were performed in R 4.1.0 [64]. GAMM models were fitted using package *mgcv* (version 1.8.-35 [70]). Post-hoc tests were carried out with the package *emmeans* (version 1.6.3 [71]). Due to the structure and high number of comparisons, only *p*-values smaller than 0.001 were considered statistically significant.

## 3. Results

### 3.1. Comparison of the Coastal Areas

The ranges of surface water temperature, salinity, water transparency (measured as Secchi depth), stratification (measured as stratification index E), and TP in the studied coastal sea areas during mid and late summer in 2009–2020 are shown in Figure 2. The sheltered AS was characterized by the highest salinity and a somewhat higher temperature compared to the other areas. The eGF had the lowest salinity of the areas, while water transparency tended to be higher and stratification slightly stronger than in the other sea areas. The wGF was set apart from the other areas by displaying a high variability in all environmental variables and by generally having the highest TP. In accordance with the TP concentrations, total phytoplankton biomass tended to be the highest in the wGF and the lowest in the BS and the eGF (Figure 2).

### 3.2. Effects of Environmental Variables on the Traits

The GAMM analyses showed a significant positive effect of surface water temperature on the biomasses and shares of N-fixing phytoplankton, buoyant phytoplankton, and harmful cyanobacteria (Table 1). With the biomass and share of harmful eukaryotic phytoplankton, temperature showed a significant non-linear (unimodal) relation. The shares of non-N-fixing, non-buoyant, and non-harmful phytoplankton were negatively affected by temperature. A negative effect of temperature was also shown on the biomass and share of motile and mixotrophic phytoplankton. On the other hand, the temperature had a positive effect on the biomass and share of autotrophic phytoplankton.

For surface water transparency, the GAMM analyses showed significant positive or negative relations to most of the response variables (Table 1). In most cases, both the biomass of phytoplankton carrying a given trait as well as the biomass of phytoplankton not carrying the trait were negatively related to water transparency. Water transparency showed a negative relation to the biomasses of both N-fixing phytoplankton and non-N-fixing phytoplankton (Table 1). However, since the share of N-fixers of total phytoplankton biomass was negatively related to transparency, while the share of non-fixers was positively related, it can be concluded that higher water transparency is more positively related to non-N-fixing phytoplankton than to N-fixing phytoplankton. The results concerning the buoyancy trait were similar, indicating that higher water transparency is more positively related to phytoplankton not carrying the trait of buoyancy than to buoyant phytoplankton.

The biomasses of both motile and non-motile phytoplankton were negatively associated with water transparency (Table 1). The share of mixotrophic phytoplankton biomass of total phytoplankton was positively related to water transparency, while autotrophic phytoplankton biomass and its share of total phytoplankton were negatively related to water transparency (Table 1). Water transparency was negatively related to the biomass of both small-sized and large-sized phytoplankton (Table 1). The biomass share of non-harmful phytoplankton of total phytoplankton biomass was positively associated with water transparency, even though the biomass of non-harmful phytoplankton was negatively associated with water transparency (Table 1). The biomass of harmful cyanobacteria, their share of total phytoplankton biomass, and the share of harmful eukaryotic phytoplankton were all negatively related to water transparency.

There was a significant negative relation between TP and the share of N-fixing, buoyant, mixotrophic, and small-sized phytoplankton, as well as with harmful cyanobacteria and harmful eukaryotic phytoplankton (Table 1). TP was positively correlated with biomass and share of non-N-fixing, non-buoyant, large-sized, and non-harmful phytoplankton, as well as with the biomass of autotrophs.

Salinity, stratification, and nutrient loading source type did not show any statistically significant effects on the phytoplankton traits that were included in our study (Table 1).

### 3.3. Regional Variability in Trait Occurrences

Based on the GAMM results, the sea area from which the phytoplankton communities originated had a significant effect on the biomass of motile phytoplankton, the biomass and share of mixotrophs, the share of autotrophs, the biomass, and share of both small- and large-sized phytoplankton, and the biomass of harmful eukaryotic phytoplankton, i.e., on 9 of the 27 studied phytoplankton variables (Table 1).

We used a post-hoc test for further pairwise comparisons between the sea areas (Table 2). Even though many traits did not show statistically significant regional differences, many of them were nearly significant, and thus, all phytoplankton variables were included in the analysis, and their results are shown in Table 2. The results showed that the biomass of motile phytoplankton was significantly lower in the BS than in the wGF and AS (Table 2, Figure 3a). The biomass of mixotrophs was significantly lower in the BS than in all other areas and also in the AS compared to the wGF (Table 2, Figure 3b). However, the share of mixotrophs was significantly lower only in the wGF compared to the BS (Table 2, Figure 3c). The share of autotrophs was significantly higher in the BS than in the wGF (Table 2, Figure 3d).

Both the biomass and share of small-sized phytoplankton were significantly higher in the AS compared to the eGF (Table 2, Figure 3e,f). In accordance with this result, the share of large-sized phytoplankton was significantly lower in the AS than in the eGF (Table 2, Figure 3g). The biomass of large-sized phytoplankton was significantly lower in the BS than in both the eastern and western Gulf of Finland (Table 2, Figure 3h). The biomass of harmful eukaryotic phytoplankton was significantly lower in the BS than in the AS and the wGF (Table 2, Figure 3i). All in all, the two areas differing the most from each other were the BS and the wGF, with a total of six differing phytoplankton trait variables. The other sea area pairs differed concerning 1–3 trait variables.

## 4. Discussion

Trait-based phytoplankton studies offer the possibility to draw conclusions beyond a single ecosystem since the results are not restricted to local taxonomical features of the community [2]. In our study, we utilized the trait-based approach with natural phytoplankton community data to gain knowledge on potential connections between the functional composition of phytoplankton and the environment. Our focus was on examining how certain phytoplankton traits (N-fixing, motility, buoyancy, mixotrophy, cell size, and harmfulness) are related to physical features connected to global change (surface layer temperature, salinity, stratification), to water quality features connected to catchment change (water transparency), and to nutrient availability connected to nutrient loading (total phosphorus, nutrient loading source type, i.e., river, point, sediment, and offshore). In addition, we took regionality into account when analyzing the data.

### 4.1. Physical Features Connected to Global Change vs. Traits

All the selected traits had a significant connection to at least one of the environmental variables included (Table 1). Of the environmental parameters connected to the Baltic Sea-wide and global trends, only surface water temperature showed a significant connection to the traits. Temperature was a major player in distinguishing the traits benefiting from its rise from those declining with potentially warmer summers (Table 1). N-fixing, buoyant, non-motile, and autotrophic phytoplankton all benefited from an increase in temperature. Harmful cyanobacteria also thrived in a warmer temperature. The positive connection between harmful cyanobacteria and temperature supports our hypothesis and is in accordance with earlier studies [72]. The result also supports that there is a connection between the increasing risk of HABs and global climate change [45], as concerns cyanobacteria. However, harmful phytoplankton other than cyanobacteria did not respond to temperature in our study. These differences between harmful cyanobacteria and other harmful phytoplankton agree with the findings of a global study [46], concluding that HAB trends need to be considered regionally and at the species level since there is no support for increasing global HAB trends that could be connected to climate change. In our data, the temperature range was quite wide (8–26 °C, Figure 2), even though only mid and late summer samples were included. The positive connection of autotrophs to temperature is explained by the nature of seasonal succession in the Baltic Sea. In summer, photosynthesis is based on remineralized nutrients, the cycling of which is enhanced by warm temperature.

Salinity did not have apparent effects on any of the traits (Table 1). This is an interesting result since salinity has been shown to markedly affect the taxonomic composition of coastal phytoplankton communities in the Baltic Sea [55,73]. The Baltic Sea phytoplankton is a mixture of marine and freshwater species [53], which share the same traits in both environments. Our results indicate a seamless trait continuum even if the salinity environment selects only species adapted to certain salinities. The result demonstrates that there is potential for developing a trait-based environmental status indicator approach, which could be applicable across salinity gradients, at least within the studied salinity range (1–6.6 psu, Figure 2).

Similar to salinity, the strength of stratification also did not have apparent effects on any of the traits (Table 1). Based on earlier studies, we hypothesized that organism size would be negatively correlated with a stronger stratification, since large cells tend to sink faster than small cells [27,28,29,74]. We also hypothesized that the potential for motility or buoyancy (i.e., the ability to stay in the phototrophic layer) would be more important when stratification is strong [48], indicating a stable water column. The reason why our results did not support these hypotheses and the respective earlier studies may be that the differences in the strength of the stratification between the four studied sea areas were not large enough. Thus, it can be concluded that within this range of stratification strength (Figure 2), stratification does not have a significant effect on the studied traits. A contributing factor worth considering is that sinking and cell size are not connected in a straightforward and constant manner [22].

### 4.2. Water Quality Features Connected to Catchment Change vs. Traits

Water transparency, measured as Secchi depth, was the environmental variable that had an effect on most of the traits in our study (Table 1). The connections between the traits and transparency were mainly negative. This may be due to transparency being strongly affected by watercolor, which also indicates the amount of dissolved organic matter. Transparency may also act as a proxy for other water quality parameters, namely N availability, which makes its direct use as a descriptor of light limitation challenging. However, since an environmental status target is to increase water transparency (Secchi depth) in the coastal waters of the Baltic Sea [75], our results support this target by showing that harmful cyanobacteria and other harmful phytoplankton are negatively correlated with water transparency. In addition, the shares of both non-harmful and larger-sized phytoplankton (the share of large-sized species and average ESD) are positively correlated with water transparency, which is relevant for more efficient grazing food chains compared to the microbial loop-based energy transfer [76].

We hypothesized that the potential for motility or buoyancy would be negatively related to water transparency. The hypothesis was based on the assumption that motility and buoyancy would be more important when water transparency is low since (in a stable water column) these traits enable cells to actively seek out the phototrophic layer [19,20,21]. We also hypothesized that the potential for mixotrophy would correlate negatively with water transparency since mixotrophs might gain a competitive advantage when water transparency is low and autotrophic growth is limited by light. This hypothesis was based on earlier findings from especially humic lake studies [49,50]. Our results did not support these hypotheses and the respective earlier studies (Table 1). One reason for this may be that the range in water transparency (0.2–8.8 m, Figure 2), together with the concurrent stratification conditions, was not great enough to show a significant effect on the traits. Another reason could be that other environmental factors affected the presence of motile and buoyant phytoplankton more than water transparency, since the same species carrying these traits also carry several other traits, which may be more important for their dynamics in the study area.

### 4.3. Nutrient Availability Connected to Nutrient Loading vs. Traits

In the mitigation of Baltic Sea eutrophication, an important target is to decrease P availability [77]. The source of the nutrient load is indicative of the quality of the incoming nutrients: the N:P ratio of the internal nutrient load from the sediments tends to be P-weighted and more important in late summer, while the N:P ratio of the river load is N-weighted and is emphasized in winter and spring. Nutrient loading from point sources is often directly utilizable for phytoplankton, and the load is constant throughout the year (municipalities) or focused on the summer (aquaculture). The loading from offshore, on the other hand, is significant only in the months before the phytoplankton spring bloom uses up the available nutrients in the surface layer, after which there are less dissolved nutrients coming from offshore areas to the coastal areas in summer. However, according to our results, the origin of the P load was not relevant for the selected phytoplankton traits since the nutrient loading source type was not significantly related to any trait (Table 1). Thus, it can be concluded that the amount of nutrient loading is more important than its origin.

The amount of TP (range 1.5–135 µg/L, Figure 2) was positively linked to the total autotrophic biomass (Table 1). It could be expected that the availability of nutrient resources enhances autotrophic growth, even though there are also studies that have shown an increase in marine phytoplankton biomass coinciding with a decreasing nutrient level [78]. TP was also significantly positively correlated to large, non-buoyant, and non-harmful phytoplankton. The observed decline of mixotrophy is a logical consequence of enhanced nutrient availability since mixotrophy is also used to supplement nutrient reserves in cells [25].

We hypothesized that the potential for N-fixing is positively correlated with TP, since N does not limit the growth of N-fixers [13]. Unexpectedly, our results did not support this hypothesis (Table 1). This finding is surprising since N-fixation should give a competitive advantage to N-fixing species, especially in situations of N limitation [12,13]. On the other hand, N-fixation in marine systems is known to be regulated not only by N limitation but also by complex interactions of abiotic and biotic factors [14]. In addition, it is known that N-fixation and N-fixing cyanobacteria may dominate in cases when N is not limiting because cyanobacteria also prefer ammonium as their inorganic N source [15]. However, our result is in accordance with an earlier study from the coastal northern Baltic Sea, thus concluding that N-fixing cyanobacteria were not directly gaining a competitive advantage as a simple function of N limitation [55]. In that study, it was concluded that the biomass of N-fixing cyanobacteria seemed to be primarily regulated by other factors than the current nutrient situation, i.e., by longer-term nutrient dynamics. This might also be an explanation for our results. Other explanations could be that the N-fixing cyanobacteria population did not actually develop at the sampling site, but had drifted there, or that the N-fixers do not solely benefit from N-fixation since other competing phytoplankton are able to take up the nutrients (P as well as the N fixed by the N-fixers) more efficiently. The latter explanation may be supported by studies showing that up to half of the fixed N can be quickly released by the cyanobacteria for the use of other microorganisms [16,17]. In addition, clear negative effects caused by N-fixing cyanobacteria on the ambient plankton taxa have not been found in studies utilizing monitoring data from the Baltic Sea [79,80].

The challenge in using the available monitoring data is that it does not cover the complete size range of phytoplankton. Single-celled <2-µm picoplankton are not included since the inverted light microscopy method is not suitable for enumerating them reliably [58]. Specifically, this fact affects all measures using phytoplankton biomass. The exclusion of picoplankton has an effect on our results since picocyanobacteria are an important part of the northern Baltic Sea phytoplankton [81,82]. This directly affects the traits concerning cell size. It has also been suggested that some of the Baltic Sea picocyanobacteria are able to fix N [83]. However, this has been challenged by a study indicating that the Baltic Sea picocyanobacteria would not fix N, but instead use ammonium released from filamentous N-fixing cyanobacteria [16]. New knowledge on N-fixation emerges, e.g., the role of symbiotic unicellular cyanobacteria in the UCYN-A/haptophyte complex may change our view of N-fixation in many environments [84,85].

In any case, our result is interesting, since the increased N-fixing cyanobacterial blooms are considered a symptom of anthropogenic eutrophication in the Baltic Sea [86,87,88]. Thus, the duration, volume, and severity of N-fixing cyanobacterial blooms have been suggested to be used as a eutrophication status indicator [89,90] in the Baltic Sea area. Our result is not necessarily contradictory to this since our study was spatially and temporally much more restricted compared to the Baltic Sea-wide multi-decadal examinations.

Our hypothesis that harmfulness would be positively correlated with nutrient resources was not supported either (Table 1). Thus, our result did not support the studies connecting an increasing risk of harmful algal blooms (HABs) to anthropogenic eutrophication [45]. Instead, this result further supports the global study concluding that it is not possible to give an all-encompassing statement regarding increasing global HAB trends connected to either climate change or nutrient pollution [46].

### 4.4. Regionality

Our four coastal study areas differed from each other by their environmental characteristics (Figure 2) and trait properties (Table 2, Figure 3). Sea areas close to each other are naturally more alike than more remote areas. For example, the two parts of the Gulf of Finland (wGF and eGF) shared similar trait properties, and the AS was most similar to the wGF. However, the two areas differing the most from each other were the BS and the wGF, which are not geographical neighbors, but they are not the two areas farthest apart either (Figure 1).

Earlier studies have shown that the taxonomic phytoplankton community composition (species-level biomass composition) is different in different coastal Baltic Sea areas [55,91]. Thus, utilizing specific taxa for environmental status indicators may be difficult. Our study shows that the trait-based approach could be used to overcome this problem. Many phytoplankton traits included in our study show clear differences between the sea areas, indicating that their development can be used for monitoring if coastal waters become more similar or dissimilar, and what are the most important drivers, if changes occur. Monitoring this development is important in order to anticipate potential changes which may lie ahead, for example, in assessing changes in the properties of pelagic food webs.

Our regionality analysis (Table 2) showed that (1) traits should be examined in both absolute terms (e.g., biomass) and in proportions (e.g., share of total biomass), and that (2) many traits do not currently show statistically significant regional differences, however, the differences are nearly significant, and thus they should be monitored. In addition, a weighted approach to monitor the development of sea areas should be considered, considering at least those traits which showed nearly significant differences between the sea areas. For example, although the biomass of non-buoyant phytoplankton did not differ significantly between the sea areas in our study (minimum *p* = 0.001, Table 2), it may be useful to include it in the analysis of how phytoplankton communities will develop in the future. A trait-based similarity analysis for the sea areas could be an informative next step, but it may require more intensive datasets.

### 4.5. Future Research Directions

Even though we do not suggest and test a new indicator concept here, one motivation behind our study was to find new perspectives to supplement and particularize the present ecological status indicators, which currently mainly rely on summarized information such as total nutrients or chlorophyll-*a*. To our knowledge, similar studies utilizing the trait-based approach have not been published. Our next aim is to develop a widely applicable trait-based phytoplankton indicator to supplement ecological status assessments. This kind of indicator development would benefit from a long-term study of phytoplankton traits to define reference conditions and suitable target values for the potential indicator. Trait-based analyses also have the potential for giving an indication of the consequences of predicted trends in environmental conditions for phytoplankton communities.

## 5. Conclusions

We used the trait-based approach in a novel way to examine how specific phytoplankton traits are related to physical features connected to global change, water quality features connected to catchment change, and nutrient availability connected to nutrient loading. The regional aspect was also considered. There is a high demand for additional information on these connections, especially for the purposes of coastal marine status assessment and revealing trends in phytoplankton communities hidden underneath simple chlorophyll-*a* values.

Our results showed that (1) of the physical features connected to global climate change, temperature was the most important affecting the presence of traits. N-fixing, buoyant, non-motile, autotrophic phytoplankton, and harmful cyanobacteria benefited from a higher temperature. Salinity and stratification did not have clear effects on any of the traits. (2) Water transparency, which is a water quality feature connected to catchment change, had a mostly negative relation to the traits. Harmfulness was negatively correlated with water transparency, while the share of non-harmful and large-sized phytoplankton were positively correlated with it. (3) The nutrient loading source type did not significantly relate to any of the traits. The potential for N-fixing was not related to TP, potentially because N-fixers do not necessarily develop at the sampling site (they may be transported from elsewhere), other phytoplankton are still able to take up nutrients more efficiently, or N-fixers are primarily regulated by other factors than the current nutrient situation, i.e., longer-term nutrient dynamics. (4) The regionality analysis showed that traits should be calculated in both absolute terms and proportions, and that even though many traits do not currently show statistically significant regional differences, several differences were nearly significant, and thus they should be monitored.

## Figures and Tables

**Figure 1 microorganisms-09-02477-f001:**
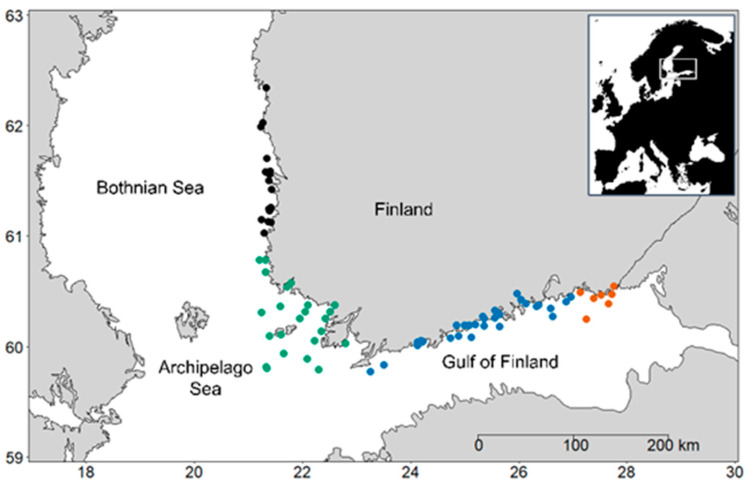
Map showing the location of the coastal sampling stations in the northern Baltic Sea. Black: Bothnian Sea (BS; 17 stations, 110 samples); green: Archipelago Sea (AS; 24 stations, 201 samples); blue: western Gulf of Finland (wGF; 31 stations, 460 samples); red: eastern Gulf of Finland (eGF; 8 stations, 141 samples).

**Figure 2 microorganisms-09-02477-f002:**
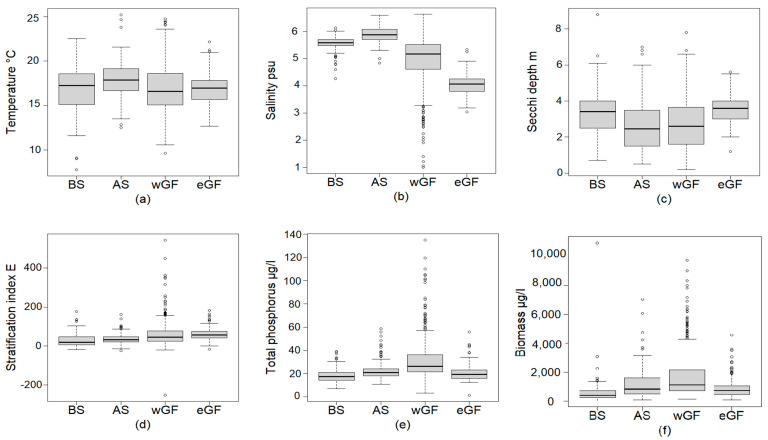
(**a**) Temperature, (**b**), salinity, (**c**) water transparency measured as Secchi depth, (**d**) stratification measured as stratification index E, (**e**) total phosphorus concentration, and (**f**) total phytoplankton biomass during mid and late summer (1 June–15 September) in 2009–2020 in the coastal sea areas (BS = Bothnian Sea; AS = Archipelago Sea; wGF = western Gulf of Finland; eGF = eastern Gulf of Finland). The bold line inside the boxes is the median, the box shows the lower and upper quartile, and the 1.5 interquartile range is shown with the whiskers. The dots outside the whiskers are extreme values.

**Figure 3 microorganisms-09-02477-f003:**
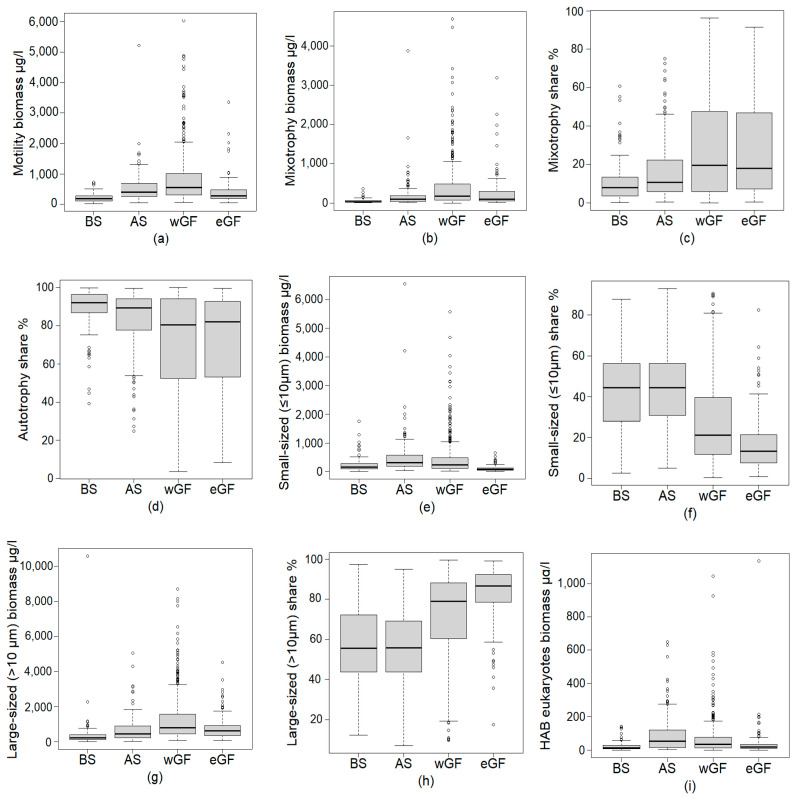
(**a**) The biomass of motile phytoplankton, (**b**) the biomass of mixotrophs, (**c**) the share of mixotrophs of total phytoplankton biomass, (**d**) the share of autotrophs of total phytoplankton biomass, (**e**) the biomass of small-sized (≤10 µm) phytoplankton, (**f**) the share of small-sized phytoplankton of total phytoplankton biomass, (**g**) the biomass of large-sized (>10 µm) phytoplankton, (**h**) the share of large-sized phytoplankton of total phytoplankton biomass, and (**i**) the biomass of harmful eukaryotic phytoplankton during mid and late summer (1 June–15 September) in 2009–2020 in the coastal sea areas (BS = Bothnian Sea; AS = Archipelago Sea; wGF = western Gulf of Finland; eGF = eastern Gulf of Finland. The bold line inside the boxes is the median, the box shows the lower and upper quartile, and the 1.5 interquartile range is shown with the whiskers. The dots outside the whiskers are extreme values.

**Table 1 microorganisms-09-02477-t001:** Results (*p*-values) of the generalized additive mixed models (GAMM), with surface temperature (Temp), surface salinity (Sal), water transparency measured as Secchi depth (Secchi), stratification index (E), total phosphorus concentration (TP), loading source type (L), and sea area (Area) as fixed factors and the hierarchical data structure (sea area > water body > sampling station) as a random factor. Statistically significant effects (*p* < 0.001) are in bold font, and colors identify significant linear positive (orange) and negative (blue) effects of the continuous explanatory variables. Significant non-linear (unimodal) effects are marked in grey color.

	Explanatory Variable						
Response Variable ^1^	Temp	Sal	Secchi	E	TP	L	Area
Nfix biom	**<0.001**	0.073	**<0.001**	0.101	0.007	0.779	0.049
Nfix share	**<0.001**	0.221	**<0.001**	0.089	**<0.001**	0.513	0.086
nonNfix biom	0.004	0.845	**<0.001**	0.008	**<0.001**	0.632	0.002
nonNfix share	**<0.001**	0.563	**<0.001**	0.167	**<0.001**	0.518	0.199
Buo biom	**<0.001**	0.193	**<0.001**	0.043	0.022	0.769	0.109
Buo share	**<0.001**	0.424	**<0.001**	0.067	**<0.001**	0.483	0.103
nonBuo biom	0.003	0.765	**<0.001**	0.006	**<0.001**	0.630	0.001
nonBuo share	**<0.001**	0.775	**<0.001**	0.120	**<0.001**	0.489	0.188
Mot biom	**<0.001**	0.606	**<0.001**	0.104	0.050	0.987	**<0.001**
Mot share	**<0.001**	0.107	**<0.001**	0.681	0.355	0.388	0.010
nonMot biom	**<0.001**	0.301	**<0.001**	0.096	0.001	0.820	0.228
nonMot share	**<0.001**	0.241	**<0.001**	0.997	0.841	0.456	0.011
MX biom	**<0.001**	0.010	0.009	0.509	0.049	0.361	**<0.001**
MX share	**<0.001**	0.156	**<0.001**	0.298	**<0.001**	0.570	**<0.001**
AU biom	**<0.001**	0.989	**<0.001**	0.021	**<0.001**	0.886	0.238
AU share	**<0.001**	0.246	**<0.001**	0.728	0.349	0.779	**<0.001**
Small biom	0.454	0.457	**<0.001**	0.016	0.842	0.842	**<0.001**
Small share	0.082	0.643	**<0.001**	0.955	**<0.001**	0.227	**<0.001**
Large biom	0.006	0.240	**<0.001**	0.089	**<0.001**	0.748	**<0.001**
Large share	0.049	0.823	**<0.001**	0.793	**<0.001**	0.179	**<0.001**
aveESD	0.001	0.128	**<0.001**	0.405	0.135	0.027	0.186
HABalg biom	**<0.001**	0.540	**<0.001**	0.047	0.002	0.470	**<0.001**
HABalg share	**<0.001**	0.809	0.001	0.003	**<0.001**	0.208	0.001
HABcyano biom	**<0.001**	0.160	**<0.001**	0.058	0.010	0.704	0.102
HABcyano share	**<0.001**	0.404	**<0.001**	0.040	**<0.001**	0.419	0.137
nonHAB biom	0.007	0.731	**<0.001**	0.004	**<0.001**	0.547	0.004
nonHAB share	**<0.001**	0.133	**<0.001**	0.088	**<0.001**	0.437	0.835

^1^ **Nfix****biom** = biomass of N-fixing phytoplankton, **Nfix share** = share of Nifx biom of total biomass, **nonNfix biom** = biomass of non-N-fixing phytoplankton, **nonNfix share** = share of nonNifx biom, **Buo biom** = biomass of buoyant phytoplankton, **Buo share** = share of Buo biom, **nonBuo biom** = biomass of non-buoyant phytoplankton, **nonBuo share** = share of nonBuo biom, **Mot biom** = biomass of motile phytoplankton, **Mot share** = share of Mot biom, **nonMot biom** = biomass of non-motile phytoplankton, **nonMot share** = share of nonMot biom, **MX biom** = biomass of mixotrophic phytoplankton, **MX share** = share of MX biom, **AU biom** = biomass of phytoplankton not carrying the trait of mixotrophy, **AU share** = share of AU biom, **Small biom** = biomass of small-sized (≤10 µm) phytoplankton, **Small share** = share of Small biom, **Large biom** = biomass of large-sized (>10 µm) phytoplankton, **Large share** = share of Large biom, **aveESD** = average Equivalent Spherical Diameter per sample, **HABalg biom** = biomass of harmful eukaryotic phytoplankton, **HABalg share** = share of HABalg biom, **HABcyano biom** = biomass of harmful cyanobacteria, **HABcyano share** = share of HABcyano biom, **nonHAB biom** = biomass of non-harmful phytoplankton, **nonHAB share** = share of nonHAB biom of total biomass.

**Table 2 microorganisms-09-02477-t002:** Results (*p*-values) of the post-hoc test for pairwise comparisons between the coastal sea areas (BS = Bothnian Sea, eGF = eastern Gulf of Finland, wGF = western Gulf of Finland, AS = Archipelago Sea). Statistically significant differences (*p* < 0.001) are in bold font.

	Sea Area Pairs					
Response Variable ^1^	BS–eGF	BS–wGF	BS–AS	eGF–wGF	eGF–AS	wGF–AS
Nfix biom	0.283	0.160	0.984	0.984	0.227	0.077
Nfix share	0.781	0.809	0.847	0.974	0.201	0.068
nonNfix biom	0.070	0.001	0.002	0.966	0.987	1.000
nonNfix share	0.946	0.950	0.798	0.996	0.411	0.159
Buo biom	0.496	0.320	0.999	0.997	0.352	0.122
Buo share	0.915	0.931	0.713	0.990	0.264	0.080
nonBuo biom	0.066	0.001	0.001	0.973	0.988	1.000
nonBuo share	0.973	0.971	0.732	0.999	0.438	0.148
Mot biom	0.014	**<0.001**	**<0.001**	0.068	0.805	0.285
Mot share	0.898	0.329	0.299	0.031	0.122	1.000
nonMot biom	0.264	0.283	0.783	0.870	0.490	0.612
nonMot share	0.858	0.433	0.241	0.037	0.060	0.965
MX biom	**<0.001**	**<0.001**	**<0.001**	0.719	0.276	**<0.001**
MX share	0.005	**<0.001**	0.002	0.652	0.746	0.005
AU biom	0.657	0.219	0.251	0.980	0.993	1.000
AU share	0.017	**<0.001**	0.033	0.713	0.565	0.001
Small biom	0.333	0.741	0.110	0.003	**<0.001**	0.234
Small share	0.002	0.121	0.993	0.033	**<0.001**	0.003
Large biom	**<0.001**	**<0.001**	0.027	0.838	0.017	0.003
Large share	0.005	0.107	0.997	0.075	**<0.001**	0.003
aveESD	0.442	0.126	0.346	0.999	0.989	0.895
HABalg biom	0.083	**<0.001**	**<0.001**	0.642	0.776	1.000
HABalg share	0.553	0.019	0.001	0.624	0.289	0.585
HABcyano biom	0.423	0.268	0.994	0.993	0.337	0.131
HABcyano share	0.890	0.926	0.785	0.980	0.279	0.113
nonHAB biom	0.081	0.003	0.006	0.989	1.000	0.986
nonHAB share	0.984	0.888	0.998	0.994	0.993	0.872

^1^ Descriptions for the response variable abbreviations are given in the footer of Table 1.

## Data Availability

Data are available in the Finnish national open database OIVA (http://www.syke.fi/en-US/Open_information, in Finnish, accessed on 22 October 2021) and in the database of the European Marine Observation and Data Network (EMODnet, https://emodnet.ec.europa.eu/en/portals accessed on 22 October 2021). The trait data table is available as Appendix A.

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
