# Peer review of "Phytoplankton Morpho-Functional Trait Variability along Coastal Environmental Gradients"

_microorganisms, 2021, doi:10.3390/microorganisms9122477_

Round 1

Reviewer 1 Report

General comments

Review microorganisms-1485457 – “Phytoplankton morpho-functional trait variability along coastal environmental gradients” by Sirpa Lehtinen et al.

The authors have done a lot of interesting work. The Authors presented original research. The study states how the research fills the identified knowledge gap. I think it's a very well-written manuscript.

From a scientific point of view, I have only two small concerns with this paper.

1) L212: Why were picocyanobacteria and Synechococcus sp. not included in the work? I hope that in the future the Authors will perform similar calculations for these organisms because their numbers and even biomass sometimes exceed other phytoplankton species. Moreover, there are reports that these species can induce toxins and fix nitrogen. I am asking the authors for a fragment of the Discussion section on the occurrence of picocyanobacteria in the Baltic Sea.

2) I am not sure but it seems to me that the citation should be in numerical order. If I am right, the citations need to be changed in the whole MS.

 In conclusion, I believe that the manuscript fits into the Journal’s aims and scope, and I think it is interesting enough to be published after a minor revision.  

Specific comments:

L262: “Prymnesiales spp.” - please correct this mistake.

L735: Please remove double spaces

Figure 2 and 3: Is it possible to ask the authors to enlarge the font on the x and y axes as it is hardly legible.

Author Response

Manuscript ID: microorganisms-1485457

Phytoplankton morpho-functional trait variability along coastal environmental gradients

Sirpa Lehtinen, Sanna Suikkanen, Heidi Hällfors, Jarno Tuimala and Harri Kuosa

Response to Reviewer 1 Comments

General comments

The authors have done a lot of interesting work. The Authors presented original research. The study states how the research fills the identified knowledge gap. I think it's a very well-written manuscript. From a scientific point of view, I have only two small concerns with this paper.

Response: We thank the Reviewer for this statement.

Point 1)

 L212: Why were picocyanobacteria and Synechococcus sp. not included in the work? I hope that in the future the Authors will perform similar calculations for these organisms because their numbers and even biomass sometimes exceed other phytoplankton species. Moreover, there are reports that these species can induce toxins and fix nitrogen. I am asking the authors for a fragment of the Discussion section on the occurrence of picocyanobacteria in the Baltic Sea.

Response: Thank you for taking up this important issue. We agree that including picocyanobacteria and Synechococcus spp. in the trait analyses is a very important task.

The reason why we had to exclude them from our data set is that the used monitoring method (Lugol-preserved samples and light-microscopy) is not suitable for reliable identification of < 2 µm -sized single phytoplankton i.e. it is not possible to separate single-celled autotrophic picoplankton from e.g. single-celled heterotrophic bacteria. Unfortunately, monitoring data collected from the study area does not include data on glutaraldehyde-preserved samples analysed with epifluorescense microscopy.

We clarified this issue by adding the following sentences to the revised MS. Changes as marked as track changes in the revised MS. Here below the references are written open, but in the revised MS they are given as numbers, as requested by the Microorganism journal.

Material and Methods L218-: “Single-celled picoplankton and Synechococcus were inconsistently enumerated. Their results were also excluded from the data set since the used monitoring method (light microscopy of Lugol’s-preserved samples (HELCOM 2021) is not suitable for reliable identification of < 2 µm -sized single phytoplankton, i.e. it is not possible to separate single-celled picoplankton from e.g. single-celled heterotrophic bacteria.”

In accordance to the Reviewer’s suggestion, we also added text on the occurrence of picocyanobacteria in the Baltic Sea in the Discussion section of the revised MS:

Discussion L665-: “The challenge in using the available monitoring data is that it does not cover the complete size range of phytoplankton. Single-celled <2-µm picoplankton are not included since the inverted light microscopy method is not suitable for enumerating them reliably (HELCOM 2021). Specifically, this fact affects all measures using phytoplankton biomass. The exclusion of picoplankton has an effect on our results, since picocyanobacteria are an important part of the northern Baltic Sea phytoplankton (Kuosa 1991, Paczkowska et al. 2020). This directly affects the traits concerning cell size. It has also been suggested that some of the Baltic Sea picocyanobacteria are able to fix N (Wasmund et al. 2001). However, this has been challenged by a study indicating that the Baltic Sea picocyanobacteria would not fix N, but instead use ammonium released from filamentous N-fixing cyanobacteria (Karlson et al. 2015). New knowledge on N-fixation emerges, and e.g. the role of symbiotic unicellular cyanobacteria in the UCYN-A/haptophyte complex may change our view of N-fixation in many environments (Martínez-Pérez 2016, Harding et al. 2018).”

We hope that in the future we can perform similar trait-based analyses including also the single-celled picoplankton, but currently this important group is unfortunately not included in the Finnish national monitoring programme.

Point 2)

I am not sure but it seems to me that the citation should be in numerical order. If I am right, the citations need to be changed in the whole MS.

Response: Yes, this is true. We changed the citations into numerical order in the revised MS.

 In conclusion, I believe that the manuscript fits into the Journal’s aims and scope, and I think it is interesting enough to be published after a minor revision. 

Response: Thank you. We have now answered all the questions asked by the Reviewer and made the requested modification in the revised MS.

Specific comments:

L262: “Prymnesiales spp.” - please correct this mistake.

Response: We corrected this in the revised MS. Thank you for pointing this out.

L735: Please remove double spaces

Response: We removed double spaces from the revised MS. Thank you for noticing this.

Figure 2 and 3: Is it possible to ask the authors to enlarge the font on the x and y axes as it is hardly legible.

Response: We enlarged the font as requested by the Reviewer.

Reviewer 2 Report

I find the manuscript very interesting, and important in the matter of environmental microbiology. Its' structure is proper, with a few sub-chapters which ease the reception of presented data. It is also written in well English, and with many updated references. Nevertheless, I have some suggestions aiming at the improvement of the manuscript:

- the units in which the salinity was measured were not specified - this should be completed

- I noticed a typo, i.e. in the word "mixotrophst" (line 543) there is no need for "t" at the end . 

Author Response

Manuscript ID: microorganisms-1485457

Phytoplankton morpho-functional trait variability along coastal environmental gradients

Sirpa Lehtinen, Sanna Suikkanen, Heidi Hällfors, Jarno Tuimala and Harri Kuosa

Response to Reviewer 2 Comments

I find the manuscript very interesting, and important in the matter of environmental microbiology. Its' structure is proper, with a few sub-chapters which ease the reception of presented data. It is also written in well English, and with many updated references.

Response: We thank the Reviewer for this statement.

Nevertheless, I have some suggestions aiming at the improvement of the manuscript:

- the units in which the salinity was measured were not specified - this should be completed

Response: We have now specified the unit in which salinity was measured (psu) in the revised MS.

- I noticed a typo, i.e. in the word "mixotrophst" (line 543) there is no need for "t" at the end .

Response: We have corrected this in the revised MS. Thank you for noticing this.

Reviewer 3 Report

The article by Lehtinen et al. entitled “Phytoplankton morpho-functional trait variability along coastal environmental gradients” examines the relationship between the phytoplankton composition and environmental factors in the northern part of the Baltic Sea. The study is based on a large set of data obtained from the analysis of 912 samples collected during the summer months of the period 2009-2020. Such trait-based approach is applied for the first time for this region. The phytoplankton morpho-functional traits included in this study are N-fixation, mixotrophy, motility, buoyancy, harmfulness and cell size. The measured environmental factors are temperature, salinity, total phosphorus concentration (TP), water transparency (Secchi depth), stratification of the water column, and nutrient loading source type. The authors build several hypotheses that test in the course of the study.

The article has its merits because it presents statistically well-supported data, enriching our knowledge about the correlations between phytoplankton morpho-functional traits and environmental factors, obtained on the basis of a method not previously applied to this area.

I have some notes:

  1. The article is quite descriptive and it is not clear what is the novelty and contribution or what is the difference compared to other studies on phytoplankton and environmental factors in the Baltic Sea. This must be emphasized.
  2. In section "Discussion", authors should compare and discuss their results with those of other authors rather than confirm or reject their own hypotheses based on their own results.
  3. The reference “GasiÅ«naitÄ—, Z.R.; Cardoso, A.C.; …. Seasonality of coastal phytoplankton in the Baltic Sea: …... Est Coast Shelf Sci, 2005, 65, 239-252.” is repeated in the reference list.

Author Response

Manuscript ID: microorganisms-1485457

Phytoplankton morpho-functional trait variability along coastal environmental gradients

Sirpa Lehtinen, Sanna Suikkanen, Heidi Hällfors, Jarno Tuimala and Harri Kuosa

Response to Reviewer 3 Comments

The article by Lehtinen et al. entitled “Phytoplankton morpho-functional trait variability along coastal environmental gradients” examines the relationship between the phytoplankton composition and environmental factors in the northern part of the Baltic Sea. The study is based on a large set of data obtained from the analysis of 912 samples collected during the summer months of the period 2009-2020. Such trait-based approach is applied for the first time for this region. The phytoplankton morpho-functional traits included in this study are N-fixation, mixotrophy, motility, buoyancy, harmfulness and cell size. The measured environmental factors are temperature, salinity, total phosphorus concentration (TP), water transparency (Secchi depth), stratification of the water column, and nutrient loading source type. The authors build several hypotheses that test in the course of the study.

The article has its merits because it presents statistically well-supported data, enriching our knowledge about the correlations between phytoplankton morpho-functional traits and environmental factors, obtained on the basis of a method not previously applied to this area.

Response: We thank the Reviewer for this statement.

I have some notes:

1. The article is quite descriptive and it is not clear what is the novelty and contribution or what is the difference compared to other studies on phytoplankton and environmental factors in the Baltic Sea. This must be emphasized.

Response: Thank you for pointing this out. We have now clarified and emphasized the novelty of our work in the revised MS. Changes as marked as track changes in the revised MS. Here below the references are written open, but in the revised MS they are given as numbers, as requested by the Microorganism journal.

Abstract L8-: “We utilized the trait-based approach in a novel way to examine how specific phytoplankton traits are related to physical features connected to global change, to water quality features connected to catchment change, and to nutrient availability connected to nutrient loading.”

Introduction L120-: “During recent years, also the trait-based approach has been introduced to the Baltic Sea phytoplankton studies by one study in which the functional phytoplankton community structure and its drivers were studied (Klais et al. 2017). The selected monitoring data set that we used in our study, has not been used for trait-based analyses earlier.”

Introduction L127-: “There is a high demand for additional information on these connections, especially for the applied purposes of coastal marine status assessments. The trait-based approach might be well suited for these purposes because it is not tied to local species composition in the same way as the taxonomy-based approach. Unlike the earlier trait-based study using Baltic Sea phytoplankton data (Klais et al. 2017), we used trait-specific biomasses and shares of total biomass in the analyses. The reason for this is that we focused specifically on certain traits, and not on the overall functional community structure. Also, our approach, i.e. using biomasses and their shares of total phytoplankton biomass, might be more feasible to be adopted in marine management.”

Conclusions L733-: “We used the trait-based approach in a novel way to examine how specific phytoplankton traits are related to physical features connected to global change, to water quality features connected to catchment change, and to nutrient availability connected to nutrient loading. The regional aspect was also considered. There is a high demand for additional information on these connections, especially for the purposes of coastal marine status assessment, and revealing trends in phytoplankton communities hidden underneath simple chlorophyll-a values.”

2. In section "Discussion", authors should compare and discuss their results with those of other authors rather than confirm or reject their own hypotheses based on their own results.

Response: As requested by the Reviewer, in the revised MS we have added more references and comparison of our results with the studies of other authors in the Discussion section:

Discussion L576-: “Based on earlier studies, we hypothesized that organism size would be negatively correlated with a stronger stratification, since large cells tend to sink faster than small cells (Kriest 2007, Finkel et al. 2010, Acevedo-Trejos et al. 2015, Sommer et al. 2016). We also hypothesized that the potential for motility or buoyancy (i.e. the ability to stay in the phototrophic layer) would be more important when stratification is strong (Ross and Sharples 2008), indicating a stable water column. The reason why our results did not support these hypotheses and the respective earlier studies may be that the differences in the strength of the stratification between the four studied sea areas were not large enough.”

Discussion L608-: “This hypothesis was based on earlier findings from especially humic lake studies (Jansson et al. 1996, Peltomaa and Ojala 2010). Our results did not support these hypotheses and the respective earlier studies (Table 1).”

Discussion L634-: “It could be expected that the availability of nutrient resources enhances autotrophic growth, even though there are also studies that have shown an increase in marine phytoplankton biomass coinciding with a decreasing nutrient level (Xu et al. 2020).”

Discussion L638-: “The observed decline of mixotrophy is a logical consequence of enhanced nutrient availability since mixotrophy is also used to supplement nutrient reserves in cells (Ward 2019).”

Discussion L645-: “This finding is surprising since N-fixation should give a competitive advantage to N-fixing species especially in situations of N limitation (Schindler 1977, Tyrrell 1999). On the other hand, N-fixation in marine systems is known to be regulated not only by N limitation but also by complex interactions of abiotic and biotic factors (Howarth & Marino 2006). In addition, it is known that N-fixation and N-fixing cyanobacteria may dominate in cases when N is not limiting because also cyanobacteria prefer ammonium as their inorganic N source (Scott & McCarthy 2010).”

Discussion L684-: “Thus, our result did not support the studies connecting an increasing risk of harmful algal blooms (HABs) to anthropogenic eutrophication (Glibert & Burford 2017).”

In addition, we added references for our hypotheses in the Introduction section:

Introduction L147-: “We hypothesized that (1) the potential for N-fixation is positively correlated with TP (since N does not limit the growth of N-fixers (Tyrrell 1999)), (2) the potential for motility or buoyancy (i.e. the ability to select for favorable growth conditions) is more important when water transparency is low and stratification is strong (Walsby et al. 1995, 1997, Smayda 1997, Ross and Sharples 2008), (3) the potential for mixotrophy correlates negatively with factors that support autotrophic growth, i.e. the availability of nutrient resources (Mitra et al. 2014) (TP in our study) and water transparency (Jansson et al. 1996, Peltomaa and Ojala 2010), (4) organism size is positively correlated with the availability of nutrient resources (since small cells are more efficient in taking up nutrients which is an advantage in nutrient-poor conditions) and negatively correlated with stratification (since large cells sink faster) (Kriest 2007, Finkel et al. 2010, Acevedo-Trejos et al. 2015), and (5) harmfulness is positively correlated with temperature and nutrient resources (i.e. global climate change and nutrient pollution (Glibert & Burford 2017)).”

3. The reference “GasiÅ«naitÄ—, Z.R.; Cardoso, A.C.; …. Seasonality of coastal phytoplankton in the Baltic Sea: …... Est Coast Shelf Sci, 2005, 65, 239-252.” is repeated in the reference list.

Response: We corrected this in the revised MS. Thank you for pointing this out.